# Trust-PCL: An Off-Policy Trust Region Method for Continuous Control

**Ofir Nachum, Mohammad Norouzi, Kelvin Xu, & Dale Schuurmans**[*]
{ofirnachum,mnorouzi,kelvinxx,schuurmans}@google.com
Google Brain

## Abstract

Trust region methods, such as TRPO, are often used to stabilize policy optimization algorithms in reinforcement learning (RL). While current trust region strategies are effective for continuous control, they typically require a large amount of on-policy interaction with the environment. To address this problem, we propose an off-policy trust region method, *Trust-PCL*, which exploits an observation that the optimal policy and state values of a maximum reward objective with a relative-entropy regularizer satisfy a set of multi-step pathwise consistencies along any path. The introduction of relative entropy regularization allows Trust-PCL to maintain optimization stability while exploiting off-policy data to improve sample efficiency. When evaluated on a number of continuous control tasks, Trust-PCL significantly improves the solution quality and sample efficiency of TRPO.[1]

## 1 Introduction

The goal of model-free reinforcement learning (RL) is to optimize an agent's behavior policy through trial and error interaction with a black box environment. Value-based RL algorithms such as Q-learning (Watkins, 1989) and policy-based algorithms such as actor-critic (Konda & Tsitsiklis, 2000) have achieved well-known successes in environments with enumerable action spaces and predictable but possibly complex dynamics, *e.g.,* as in Atari games (Mnih et al., 2013; Van Hasselt et al., 2016; Mnih et al., 2016). However, when applied to environments with more sophisticated action spaces and dynamics (*e.g.,* continuous control and robotics), success has been far more limited.

In an attempt to improve the applicability of Q-learning to continuous control, Silver et al. (2014) and Lillicrap et al. (2015) developed an off-policy algorithm DDPG, leading to promising results on continuous control environments. That said, current off-policy methods including DDPG often improve data efficiency at the cost of optimization stability. The behaviour of DDPG is known to be highly dependent on hyperparameter selection and initialization (Metz et al., 2017); even when using optimal hyperparameters, individual training runs can display highly varying outcomes.

On the other hand, in an attempt to improve the stability and convergence speed of policy-based RL methods, Kakade (2002) developed a natural policy gradient algorithm based on Amari (1998), which subsequently led to the development of trust region policy optimization (TRPO) (Schulman et al., 2015). TRPO has shown strong empirical performance on difficult continuous control tasks often outperforming value-based methods like DDPG. However, a major drawback is that such methods are not able to exploit off-policy data and thus require a large amount of on-policy interaction with the environment, making them impractical for solving challenging real-world problems.

Efforts at combining the stability of trust region policy-based methods with the sample efficiency of value-based methods have focused on using off-policy data to better train a value estimate, which can be used as a control variate for variance reduction (Gu et al., 2017a;b).

In this paper, we investigate an alternative approach to improving the sample efficiency of trust region policy-based RL methods. We exploit the key fact that, under entropy regularization, the

---

[*]Also at the Department of Computing Science, University of Alberta, daes@ualberta.ca

[1]An implementation of Trust-PCL is available at https://github.com/tensorflow/models/tree/master/research/pcl_rl

optimal policy and value function satisfy a set of pathwise consistency properties along *any* sampled path (Nachum et al., 2017), which allows both on and off-policy data to be incorporated in an actor-critic algorithm, PCL. The original PCL algorithm optimized an entropy regularized maximum reward objective and was evaluated on relatively simple tasks. Here we extend the ideas of PCL to achieve strong results on standard, challenging continuous control benchmarks. The main observation is that by alternatively augmenting the maximum reward objective with a relative entropy regularizer, the optimal policy and values still satisfy a certain set of pathwise consistencies along any sampled trajectory. The resulting objective is equivalent to maximizing expected reward subject to a penalty-based constraint on divergence from a reference (*i.e.,* previous) policy.

We exploit this observation to propose a new off-policy trust region algorithm, *Trust-PCL*, that is able to exploit off-policy data to train policy and value estimates. Moreover, we present a simple method for determining the coefficient on the relative entropy regularizer to remain agnostic to reward scale, hence ameliorating the task of hyperparameter tuning. We find that the incorporation of a relative entropy regularizer is crucial for good and stable performance. We evaluate Trust-PCL against TRPO, and observe that Trust-PCL is able to solve difficult continuous control tasks, while improving the performance of TRPO both in terms of the final reward achieved as well as sample-efficiency.

## 2 RELATED WORK

**Trust Region Methods.** Gradient descent is the predominant optimization method for neural networks. A gradient descent step is equivalent to solving a trust region constrained optimization,

$$\text{minimize } \ell(\theta + \mathrm{d}\theta) \approx \ell(\theta) + \nabla\ell(\theta)^\mathsf{T}\mathrm{d}\theta \quad \text{s.t.} \quad \mathrm{d}\theta^\mathsf{T}\mathrm{d}\theta \leq \epsilon \,, \tag{1}$$

which yields the locally optimal update $\mathrm{d}\theta = -\eta\nabla\ell(\theta)$ such that $\eta = \sqrt{\epsilon}/\|\nabla\ell(\theta)\|$; hence by considering a Euclidean ball, gradient descent assumes the parameters lie in a Euclidean space.

However, in machine learning, particularly in the context of multi-layer neural network training, Euclidean geometry is not necessarily the best way to characterize proximity in parameter space. It is often more effective to define an appropriate Riemannian metric that respects the loss surface (Amari, 2012), which allows much steeper descent directions to be identified within a local neighborhood (*e.g.,* Amari (1998); Martens & Grosse (2015)). Whenever the loss is defined in terms of a Bregman divergence between an (unknown) optimal parameter $\theta^*$ and model parameter $\theta$, *i.e.,* $\ell(\theta) \equiv D_\mathrm{F}(\theta^*, \theta)$, it is natural to use the same divergence to form the trust region:

$$\text{minimize } D_\mathrm{F}(\theta^*, \theta + \mathrm{d}\theta) \quad \text{s.t.} \quad D_\mathrm{F}(\theta, \theta + \mathrm{d}\theta) \leq \epsilon \,. \tag{2}$$

The natural gradient (Amari, 1998) is a generalization of gradient descent where the Fisher information matrix $F(\theta)$ is used to define the local geometry of the parameter space around $\theta$. If a parameter update is constrained by $\mathrm{d}\theta^\mathsf{T}F(\theta)\mathrm{d}\theta \leq \epsilon$, a descent direction of $\mathrm{d}\theta \equiv -\eta F(\theta)^{-1}\nabla\ell(\theta)$ is obtained. This geometry is especially effective for optimizing the log-likelihood of a conditional probabilistic model, where the objective is in fact the KL divergence $D_\mathrm{KL}(\theta^*, \theta)$. The local optimization is,

$$\text{minimize } D_\mathrm{KL}(\theta^*, \theta + \mathrm{d}\theta) \quad \text{s.t.} \quad D_\mathrm{KL}(\theta, \theta + \mathrm{d}\theta) \approx \mathrm{d}\theta^\mathsf{T}F(\theta)\mathrm{d}\theta \leq \epsilon \,. \tag{3}$$

Thus, natural gradient approximates the trust region by $D_\mathrm{KL}(a, b) \approx (a - b)^\mathsf{T}F(a)(a - b)$, which is accurate up to a second order Taylor approximation. Previous work (Kakade, 2002; Bagnell & Schneider, 2003; Peters & Schaal, 2008; Schulman et al., 2015) has applied natural gradient to policy optimization, locally improving expected reward subject to variants of $\mathrm{d}\theta^\mathsf{T}F(\theta)\mathrm{d}\theta \leq \epsilon$. Recently, TRPO (Schulman et al., 2015; 2016) has achieved state-of-the-art results in continuous control by adding several approximations to the natural gradient to make nonlinear policy optimization feasible.

Another approach to trust region optimization is given by proximal gradient methods (Parikh et al., 2014). The class of proximal gradient methods most similar to our work are those that replace the hard constraint in (2) with a penalty added to the objective. These techniques have recently become popular in RL (Wang et al., 2016; Heess et al., 2017; Schulman et al., 2017b), although in terms of final reward performance on continuous control benchmarks, TRPO is still considered to be the state-of-the-art.

Norouzi et al. (2016) make the observation that *entropy regularized* expected reward may be expressed as a reversed KL divergence $D_{\mathrm{KL}}(\theta, \theta^*)$, which suggests that an alternative to the constraint in (3) should be used when such regularization is present:

$$\text{minimize } D_{\mathrm{KL}}(\theta + \mathrm{d}\theta, \theta^*) \quad \text{s.t.} \quad D_{\mathrm{KL}}(\theta + \mathrm{d}\theta, \theta) \approx \mathrm{d}\theta^\mathsf{T} F(\theta + \mathrm{d}\theta)\mathrm{d}\theta \le \epsilon . \tag{4}$$

Unfortunately, this update requires computing the Fisher matrix at the endpoint of the update. The use of $F(\theta)$ in previous work can be considered to be an approximation when entropy regularization is present, but it is not ideal, particularly if $\mathrm{d}\theta$ is large. In this paper, by contrast, we demonstrate that the optimal $\mathrm{d}\theta$ under the reverse KL constraint $D_{\mathrm{KL}}(\theta + \mathrm{d}\theta, \theta) \le \epsilon$ can indeed be characterized. Defining the constraint in this way appears to be more natural and effective than that of TRPO.

**Softmax Consistency.** To comply with the information geometry over policy parameters, previous work has used the relative entropy (*i.e.,* KL divergence) to regularize policy optimization; resulting in a softmax relationship between the optimal policy and state values (Peters et al., 2010; Azar et al., 2012; 2011; Fox et al., 2016; Rawlik et al., 2013) under single-step rollouts. Our work is unique in that we leverage consistencies over multi-step rollouts.

The existence of multi-step softmax consistencies has been noted by prior work—first by Nachum et al. (2017) in the presence of entropy regularization. The existence of the same consistencies with relative entropy has been noted by Schulman et al. (2017a). Our work presents multi-step consistency relations for a hybrid relative entropy plus entropy regularized expected reward objective, interpreting relative entropy regularization as a trust region constraint. This work is also distinct from prior work in that the coefficient of relative entropy can be automatically determined, which we have found to be especially crucial in cases where the reward distribution changes dramatically during training.

Most previous work on softmax consistency (*e.g.,* Fox et al. (2016); Azar et al. (2012); Nachum et al. (2017)) have only been evaluated on relatively simple tasks, including grid-world and discrete algorithmic environments. Rawlik et al. (2013) conducted evaluations on simple variants of the CartPole and Pendulum continuous control tasks. More recently, Haarnoja et al. (2017) showed that soft Q-learning (a single-step special case of PCL) can succeed on more challenging environments, such as a variant of the Swimmer task we consider below. By contrast, this paper presents a successful application of the softmax consistency concept to difficult and standard continuous-control benchmarks, resulting in performance that is competitive with and in some cases beats the state-of-the-art.

## 3 NOTATION & BACKGROUND

We model an agent's behavior by a policy distribution $\pi(a \,|\, s)$ over a set of actions (possibly discrete or continuous). At iteration $t$, the agent encounters a state $s_t$ and performs an action $a_t$ sampled from $\pi(a \mid s_t)$. The environment then returns a scalar reward $r_t \sim r(s_t, a_t)$ and transitions to the next state $s_{t+1} \sim \rho(s_t, a_t)$. When formulating expectations over actions, rewards, and state transitions we will often omit the sampling distributions, $\pi$, $r$, and $\rho$, respectively.

**Maximizing Expected Reward.** The standard objective in RL is to maximize expected future discounted reward. We formulate this objective on a per-state basis recursively as

$$O_{\mathrm{ER}}(s, \pi) = \mathbb{E}_{a,r,s'} \left[ r + \gamma O_{\mathrm{ER}}(s', \pi) \right] . \tag{5}$$

The overall, state-agnostic objective is the expected per-state objective when states are sampled from interactions with the environment:

$$O_{\mathrm{ER}}(\pi) = \mathbb{E}_s[O_{\mathrm{ER}}(s, \pi)]. \tag{6}$$

Most policy-based algorithms, including REINFORCE (Williams & Peng, 1991) and actor-critic (Konda & Tsitsiklis, 2000), aim to optimize $O_{\mathrm{ER}}$ given a parameterized policy.

**Path Consistency Learning (PCL).** Inspired by Williams & Peng (1991), Nachum et al. (2017) augment the objective $O_{\mathrm{ER}}$ in (5) with a discounted entropy regularizer to derive an objective,

$$O_{\mathrm{ENT}}(s, \pi) = O_{\mathrm{ER}}(s, \pi) + \tau \mathbb{H}(s, \pi) , \tag{7}$$

where $\tau \ge 0$ is a user-specified temperature parameter that controls the degree of entropy regularization, and the discounted entropy $\mathbb{H}(s, \pi)$ is recursively defined as

$$\mathbb{H}(s, \pi) = \mathbb{E}_{a,s'}[- \log \pi(a \mid s) + \gamma \mathbb{H}(s', \pi)] . \tag{8}$$

Note that the objective $O_{\text{ENT}}(s, \pi)$ can then be re-expressed recursively as,

$$O_{\text{ENT}}(s, \pi) = \mathbb{E}_{a,r,s'}[r - \tau \log \pi(a \mid s) + \gamma O_{\text{ENT}}(s', \pi)] \,. \tag{9}$$

Nachum et al. (2017) show that the optimal policy $\pi^*$ for $O_{\text{ENT}}$ and $V^*(s) = O_{\text{ENT}}(s, \pi^*)$ mutually satisfy a softmax temporal consistency constraint along any sequence of states $s_0, \ldots, s_d$ starting at $s_0$ and a corresponding sequence of actions $a_0, \ldots, a_{d-1}$:

$$V^*(s_0) = \mathbb{E}_{r_i, s_i} \left[ \gamma^d V^*(s_d) + \sum_{i=0}^{d-1} \gamma^i (r_i - \tau \log \pi^*(a_i | s_i)) \right] \,. \tag{10}$$

This observation led to the development of the PCL algorithm, which attempts to minimize squared error between the LHS and RHS of (10) to simultaneously optimize parameterized $\pi_\theta$ and $V_\phi$. Importantly, PCL is applicable to both on-policy and off-policy trajectories.

**Trust Region Policy Optimization (TRPO).** As noted, standard policy-based algorithms for maximizing $O_{\text{ER}}$ can be unstable and require small learning rates for training. To alleviate this issue, Schulman et al. (2015) proposed to perform an iterative trust region optimization to maximize $O_{\text{ER}}$. At each step, a prior policy $\tilde{\pi}$ is used to sample a large batch of trajectories, then $\pi$ is subsequently optimized to maximize $O_{\text{ER}}$ while remaining within a constraint defined by the average per-state KL-divergence with $\tilde{\pi}$. That is, at each iteration TRPO solves the constrained optimization problem,

$$\underset{\pi}{\text{maximize}} \; O_{\text{ER}}(\pi) \quad \text{s.t.} \quad \mathbb{E}_{s \sim \tilde{\pi}, \rho}[\, \text{KL}\,(\tilde{\pi}(-|s) \,\|\, \pi(-|s))\,] \leq \epsilon. \tag{11}$$

The prior policy is then replaced with the new policy $\pi$, and the process is repeated.

## 4 METHOD

To enable more stable training and better exploit the natural information geometry of the parameter space, we propose to augment the entropy regularized expected reward objective $O_{\text{ENT}}$ in (7) with a discounted relative entropy trust region around a prior policy $\tilde{\pi}$,

$$\underset{\pi}{\text{maximize}} \; \mathbb{E}_s[O_{\text{ENT}}(\pi)] \quad \text{s.t.} \quad \mathbb{E}_s[\mathbb{G}(s, \pi, \tilde{\pi})] \leq \epsilon \,, \tag{12}$$

where the discounted relative entropy is recursively defined as

$$\mathbb{G}(s, \pi, \tilde{\pi}) = \mathbb{E}_{a,s'} \left[ \log \pi(a|s) - \log \tilde{\pi}(a|s) + \gamma \mathbb{G}(s', \pi, \tilde{\pi}) \right] \,. \tag{13}$$

This objective attempts to maximize entropy regularized expected reward while maintaining natural proximity to the previous policy. Although previous work has separately proposed to use relative entropy and entropy regularization, we find that the two components serve different purposes, each of which is beneficial: entropy regularization helps improve exploration, while the relative entropy improves stability and allows for a faster learning rate. This combination is a key novelty.

Using the method of Lagrange multipliers, we cast the constrained optimization problem in (13) into maximization of the following objective,

$$O_{\text{RELENT}}(s, \pi) = O_{\text{ENT}}(s, \pi) - \lambda \mathbb{G}(s, \pi, \tilde{\pi}) \,. \tag{14}$$

Again, the environment-wide objective is the expected per-state objective when states are sampled from interactions with the environment,

$$O_{\text{RELENT}}(\pi) = \mathbb{E}_s[O_{\text{RELENT}}(s, \pi)]. \tag{15}$$

### 4.1 PATH CONSISTENCY WITH RELATIVE ENTROPY

A key technical observation is that the $O_{\text{RELENT}}$ objective has a similar decomposition structure to $O_{\text{ENT}}$, and one can cast $O_{\text{RELENT}}$ as an entropy regularized expected reward objective with a set of transformed rewards, *i.e.,*

$$O_{\text{RELENT}}(s, \pi) = \widetilde{O}_{\text{ER}}(s, \pi) + (\tau + \lambda)\mathbb{H}(s, \pi), \tag{16}$$

where $\widetilde{O}_{\text{ER}}(s, \pi)$ is an expected reward objective on a transformed reward distribution function $\tilde{r}(s, a) = r(s, a) + \lambda \log \tilde{\pi}(a|s)$. Thus, in what follows, we derive a corresponding form of the multi-step path consistency in (10).

Let $\pi^*$ denote the optimal policy, defined as $\pi^* = \text{argmax}_\pi O_{\text{RELENT}}(\pi)$. As in PCL (Nachum et al., 2017), this optimal policy may be expressed as

$$\pi^*(a_t|s_t) = \exp\left\{\frac{\mathbb{E}_{\tilde{r}_t \sim \tilde{r}(s_t, a_t), s_{t+1}}[\tilde{r}_t + \gamma V^*(s_{t+1})] - V^*(s_t)}{\tau + \lambda}\right\}, \tag{17}$$

where $V^*$ are the softmax state values defined recursively as

$$V^*(s_t) = (\tau + \lambda) \log \int_{\mathcal{A}} \exp\left\{\frac{\mathbb{E}_{\tilde{r}_t \sim \tilde{r}(s_t, a), s_{t+1}}[\tilde{r}_t + \gamma V^*(s_{t+1})]}{\tau + \lambda}\right\} \mathrm{d}a. \tag{18}$$

We may re-arrange (17) to yield

$$\begin{aligned} V^*(s_t) &= \mathbb{E}_{\tilde{r}_t \sim \tilde{r}(s_t, a_t), s_{t+1}}[\tilde{r}_t - (\tau + \lambda) \log \pi^*(a_t|s_t) + \gamma V^*(s_{t+1})] &\text{(19)} \\ &= \mathbb{E}_{r_t, s_{t+1}}[r_t - (\tau + \lambda) \log \pi^*(a_t|s_t) + \lambda \log \tilde{\pi}(a_{t+i}|s_{t+i}) + \gamma V^*(s_{t+1})]. &\text{(20)} \end{aligned}$$

This is a single-step temporal consistency which may be extended to multiple steps by further expanding $V^*(s_{t+1})$ on the RHS using the same identity. Thus, in general we have the following softmax temporal consistency constraint along any sequence of states defined by a starting state $s_t$ and a sequence of actions $a_t, \ldots, a_{t+d-1}$:

$$V^*(s_t) = \mathop{\mathbb{E}}_{r_{t+i}, s_{t+i}}\left[\gamma^d V^*(s_{t+d}) + \sum_{i=0}^{d-1} \gamma^i \left(r_{t+i} - (\tau + \lambda) \log \pi^*(a_{t+i}|s_{t+i}) + \lambda \log \tilde{\pi}(a_{t+i}|s_{t+i})\right)\right]. \tag{21}$$

## 4.2 TRUST-PCL

We propose to train a parameterized policy $\pi_\theta$ and value estimate $V_\phi$ to satisfy the multi-step consistencies in (21). Thus, we define a consistency error for a sequence of states, actions, and rewards $s_{t:t+d} \equiv (s_t, a_t, r_t, \ldots, s_{t+d-1}, a_{t+d-1}, r_{t+d-1}, s_{t+d})$ sampled from the environment as

$$\begin{aligned} C(s_{t:t+d}, \theta, \phi) &= -V_\phi(s_t) + \gamma^d V_\phi(s_{t+d}) + \\ & \sum_{i=0}^{d-1} \gamma^i \left(r_{t+i} - (\tau + \lambda) \log \pi_\theta(a_{t+i}|s_{t+i}) + \lambda \log \pi_{\tilde{\theta}}(a_{t+i}|s_{t+i})\right). \end{aligned} \tag{22}$$

We aim to minimize the squared consistency error on every sub-trajectory of length $d$. That is, the loss for a given batch of episodes (or sub-episodes) $S = \{s_{0:T_k}^{(k)}\}_{k=1}^B$ is

$$\mathcal{L}(S, \theta, \phi) = \sum_{k=1}^{B} \sum_{t=0}^{T_k - 1} C(s_{t:t+d}^{(k)}, \theta, \phi)^2. \tag{23}$$

We perform gradient descent on $\theta$ and $\phi$ to minimize this loss. In practice, we have found that it is beneficial to learn the parameter $\phi$ at least as fast as $\theta$, and accordingly, given a mini-batch of episodes we perform a single gradient update on $\theta$ and possibly multiple gradient updates on $\phi$ (see Appendix for details).

In principle, the mini-batch $S$ may be taken from either on-policy or off-policy trajectories. In our implementation, we utilized a replay buffer prioritized by recency. As episodes (or sub-episodes) are sampled from the environment they are placed in a replay buffer and a priority $p(s_{0:T})$ is given to a trajectory $s_{0:T}$ equivalent to the current training step. Then, to sample a batch for training, $B$ episodes are sampled from the replay buffer proportional to exponentiated priority $\exp\{\beta p(s_{0:T})\}$ for some hyperparameter $\beta \geq 0$.

For the prior policy $\pi_{\tilde{\theta}}$, we use a lagged geometric mean of the parameters. At each training step, we update $\tilde{\theta} \leftarrow \alpha\tilde{\theta} + (1 - \alpha)\theta$. Thus on average our training scheme attempts to maximize entropy regularized expected reward while penalizing divergence from a policy roughly $1/(1 - \alpha)$ training steps in the past.

### 4.3 Automatic Tuning of The Lagrange Multiplier $\lambda$

The use of a relative entropy regularizer as a penalty rather than a constraint introduces several difficulties. The hyperparameter $\lambda$ must necessarily adapt to the distribution of rewards. Thus, $\lambda$ must be tuned not only to each environment but also during training on a single environment, since the observed reward distribution changes as the agent's behavior policy improves. Using a constraint form of the regularizer is more desirable, and others have advocated its use in practice (Schulman et al., 2015) specifically to robustly allow larger updates during training.

To this end, we propose to redirect the hyperparameter tuning from $\lambda$ to $\epsilon$. Specifically, we present a method which, given a desired hard constraint on the relative entropy defined by $\epsilon$, approximates the equivalent penalty coefficient $\lambda(\epsilon)$. This is a key novelty of our work and is distinct from previous attempts at automatically tuning a regularizing coefficient, which iteratively increase and decrease the coefficient based on observed training behavior (Schulman et al., 2017b; Heess et al., 2017).

We restrict our analysis to the undiscounted setting $\gamma = 1$ with entropy regularizer $\tau = 0$. Additionally, we assume deterministic, finite-horizon environment dynamics. An additional assumption we make is that the expected KL-divergence over states is well-approximated by the KL-divergence starting from the unique initial state $s_0$. Although in our experiments these restrictive assumptions are not met, we still found our method to perform well for adapting $\lambda$ during training.

In this setting the optimal policy of (14) is proportional to exponentiated scaled reward. Specifically, for a full episode $s_{0:T} = (s_0, a_0, r_0, \ldots, s_{T-1}, a_{T-1}, r_{T-1}, s_T)$, we have

$$\pi^*(s_{0:T}) \propto \tilde{\pi}(s_{0:T}) \exp\left\{ \frac{R(s_{0:T})}{\lambda} \right\}, \tag{24}$$

where $\pi(s_{0:T}) = \prod_{i=0}^{T-1} \pi(a_i | s_i)$ and $R(s_{0:T}) = \sum_{i=0}^{T-1} r_i$. The normalization factor of $\pi^*$ is

$$Z = \mathbb{E}_{s_{0:T} \sim \tilde{\pi}} \left[ \exp\left\{ \frac{R(s_{0:T})}{\lambda} \right\} \right]. \tag{25}$$

We would like to approximate the trajectory-wide KL-divergence between $\pi^*$ and $\tilde{\pi}$. We may express the KL-divergence analytically:

$$KL(\pi^* || \tilde{\pi}) = \mathbb{E}_{s_{0:T} \sim \pi^*} \left[ \log\left( \frac{\pi^*(s_{0:T})}{\tilde{\pi}(s_{0:T})} \right) \right] \tag{26}$$

$$= \mathbb{E}_{s_{0:T} \sim \pi^*} \left[ \frac{R(s_{0:T})}{\lambda} - \log Z \right] \tag{27}$$

$$= -\log Z + \mathbb{E}_{s_{0:T} \sim \tilde{\pi}} \left[ \frac{R(s_{0:T})}{\lambda} \cdot \frac{\pi^*(s_{0:T})}{\tilde{\pi}(s_{0:T})} \right] \tag{28}$$

$$= -\log Z + \mathbb{E}_{s_{0:T} \sim \tilde{\pi}} \left[ \frac{R(s_{0:T})}{\lambda} \exp\{R(s_{0:T})/\lambda - \log Z\} \right]. \tag{29}$$

Since all expectations are with respect to $\tilde{\pi}$, this quantity is tractable to approximate given episodes sampled from $\tilde{\pi}$

Therefore, in Trust-PCL, given a set of episodes sampled from the prior policy $\pi_{\tilde{\theta}}$ and a desired maximum divergence $\epsilon$, we can perform a simple line search to find a suitable $\lambda(\epsilon)$ which yields $KL(\pi^* || \pi_{\tilde{\theta}})$ as close as possible to $\epsilon$.

The preceding analysis provided a method to determine $\lambda(\epsilon)$ given a desired maximum divergence $\epsilon$. However, there is still a question of whether $\epsilon$ should change during training. Indeed, as episodes may possibly increase in length, $KL(\pi^* || \tilde{\pi})$ naturally increases when compared to the average per-state $KL(\pi^*(-|s) || \tilde{\pi}(-|s))$, and vice versa for decreasing length. Thus, in practice, given an $\epsilon$ and a set of sampled episodes $S = \{s_{0:T_k}^{(k)}\}_{k=1}^N$, we approximate the best $\lambda$ which yields a maximum divergence of $\frac{\epsilon}{N} \sum_{k=1}^N T_k$. This makes it so that $\epsilon$ corresponds more to a constraint on the length-averaged KL-divergence.

To avoid incurring a prohibitively large number of interactions with the environment for each parameter update, in practice we use the last 100 episodes as the set of sampled episodes $S$. While

this is not exactly the same as sampling episodes from $\pi_{\tilde{\theta}}$, it is not too far off since $\pi_{\tilde{\theta}}$ is a lagged version of the online policy $\pi_{\theta}$. Moreover, we observed this protocol to work well in practice. A more sophisticated and accurate protocol may be derived by weighting the episodes according to the importance weights corresponding to their true sampling distribution.

# 5 EXPERIMENTS

We evaluate Trust-PCL against TRPO on a number of benchmark tasks. We choose TRPO as a baseline since it is a standard algorithm known to achieve state-of-the-art performance on the continuous control tasks we consider (see *e.g.,* leaderboard results on the OpenAI Gym website (Brockman et al., 2016)). We find that Trust-PCL can match or improve upon TRPO's performance in terms of both average reward and sample efficiency.

## 5.1 SETUP

We chose a number of control tasks available from OpenAI Gym (Brockman et al., 2016). The first task, Acrobot, is a discrete-control task, while the remaining tasks (HalfCheetah, Swimmer, Hopper, Walker2d, and Ant) are well-known continuous-control tasks utilizing the MuJoCo environment (Todorov et al., 2012).

For TRPO we trained using batches of $Q = 25,000$ steps ($12,500$ for Acrobot), which is the approximate batch size used by other implementations (Duan et al., 2016; Schulman, 2017). Thus, at each training iteration, TRPO samples $25,000$ steps using the policy $\pi_{\tilde{\theta}}$ and then takes a single step within a KL-ball to yield a new $\pi_{\theta}$.

Trust-PCL is off-policy, so to evaluate its performance we alternate between collecting experience and training on batches of experience sampled from the replay buffer. Specifically, we alternate between collecting $P = 10$ steps from the environment and performing a single gradient step based on a batch of size $Q = 64$ sub-episodes of length $P$ from the replay buffer, with a recency weight of $\beta = 0.001$ on the sampling distribution of the replay buffer. To maintain stability we use $\alpha = 0.99$ and we modified the loss from squared loss to Huber loss on the consistency error. Since our policy is parameterized by a unimodal Gaussian, it is impossible for it to satisfy all path consistencies, and so we found this crucial for stability.

For each of the variants and for each environment, we performed a hyperparameter search to find the best hyperparameters. The plots presented here show the reward achieved during training on the best hyperparameters averaged over the best $4$ seeds of $5$ randomly seeded training runs. Note that this reward is based on greedy actions (rather than random sampling).

Experiments were performed using Tensorflow (Abadi et al., 2016). Although each training step of Trust-PCL (a simple gradient step) is considerably faster than TRPO, we found that this does not have an overall effect on the run time of our implementation, due to a combination of the fact that each environment step is used in multiple training steps of Trust-PCL and that a majority of the run time is spent interacting with the environment. A detailed description of our implementation and hyperparameter search is available in the Appendix.

## 5.2 RESULTS

We present the reward over training of Trust-PCL and TRPO in Figure 1. We find that Trust-PCL can match or beat the performance of TRPO across all environments in terms of both final reward and sample efficiency. These results are especially significant on the harder tasks (Walker2d and Ant). We additionally present our results compared to other published results in Table 1. We find that even when comparing across different implementations, Trust-PCL can match or beat the state-of-the-art.

### 5.2.1 HYPERPARAMETER ANALYSIS

The most important hyperparameter in our method is $\epsilon$, which determines the size of the trust region and thus has a critical role in the stability of the algorithm. To showcase this effect, we present the reward during training for several different values of $\epsilon$ in Figure 2. As $\epsilon$ increases, instability increases as well, eventually having an adverse effect on the agent's ability to achieve optimal reward.

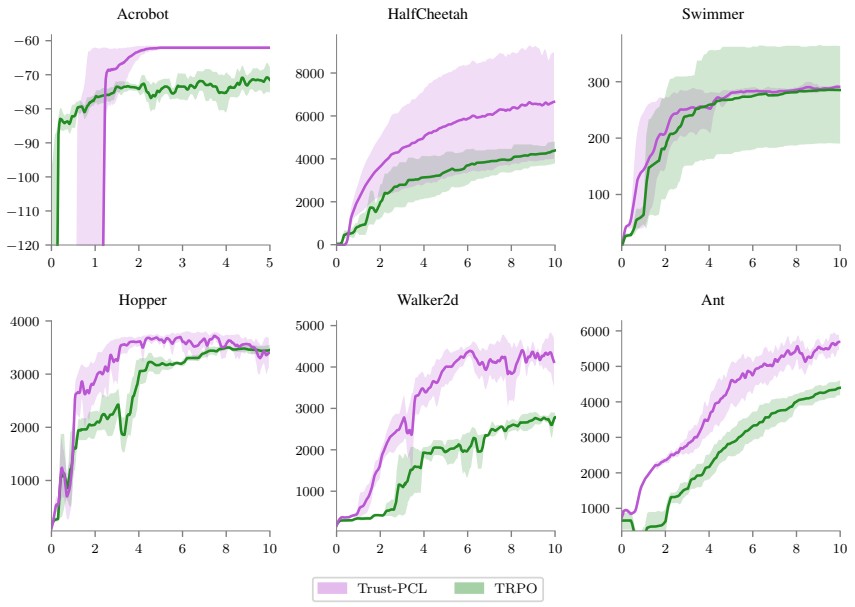

Figure 1: The results of Trust-PCL against a TRPO baseline. Each plot shows average greedy reward with single standard deviation error intervals capped at the min and max across 4 best of 5 randomly seeded training runs after choosing best hyperparameters. The x-axis shows millions of environment steps. We observe that Trust-PCL is consistently able to match and, in many cases, beat TRPO's performance both in terms of reward and sample efficiency.

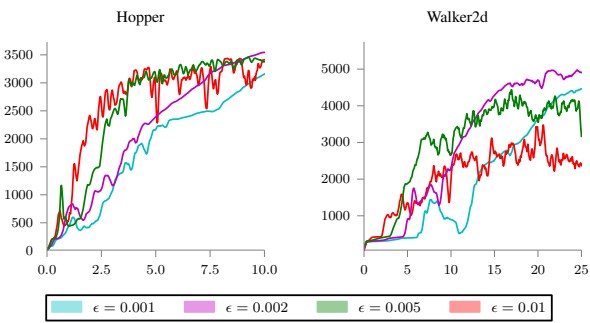

Figure 2: The results of Trust-PCL across several values of $\epsilon$, defining the size of the trust region. Each plot shows average greedy reward across 4 best of 5 randomly seeded training runs after choosing best hyperparameters. The x-axis shows millions of environment steps. We observe that instability increases with $\epsilon$, thus concluding that the use of trust region is crucial.

Note that standard PCL (Nachum et al., 2017) corresponds to $\epsilon \to \infty$ (that is, $\lambda = 0$). Therefore, standard PCL would fail in these environments, and the use of trust region is crucial.

The main advantage of Trust-PCL over existing trust region methods for continuous control is its ability to learn in an off-policy manner. The degree to which Trust-PCL is off-policy is determined by a combination of the hyparparameters $\alpha$, $\beta$, and $P$. To evaluate the importance of training off-policy, we evaluate Trust-PCL with a hyperparameter setting that is more on-policy. We set $\alpha = 0.95$, $\beta = 0.1$, and $P = 1,000$. In this setting, we also use large batches of $Q = 25$ episodes of length $P$ (a total of $25,000$ environment steps per batch). Figure 3 shows the results of Trust-PCL with our original parameters and this new setting. We note a dramatic advantage in sample efficiency when using off-policy training. Although Trust-PCL (on-policy) can achieve state-of-the-art reward performance, it requires an exorbitant amount of experience. On the other hand, Trust-PCL (off-

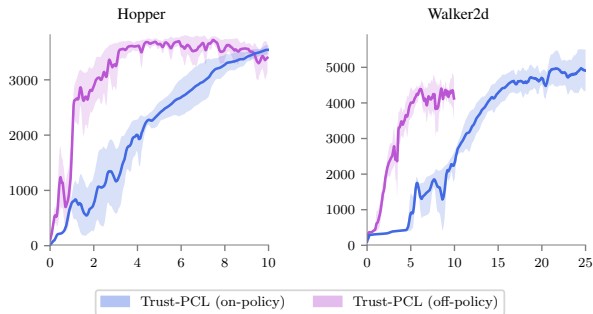

Figure 3: The results of Trust-PCL varying the degree of on/off-policy. We see that Trust-PCL (on-policy) has a behavior similar to TRPO, achieving good final reward but requiring an exorbitant number of experience collection. When collecting less experience per training step in Trust-PCL (off-policy), we are able to improve sample efficiency while still achieving a competitive final reward.

| Domain | TRPO-GAE | TRPO (rllab) | TRPO (ours) | Trust-PCL | IPG |
|---|---|---|---|---|---|
| HalfCheetah | 4871.36 | 2889 | 4343.6 | **7057.1** | 4767 |
| Swimmer | 137.25 | – | 288.1 | **297.0** | – |
| Hopper | 3765.78 | – | 3516.7 | **3804.9** | – |
| Walker2d | **6028.73** | 1487 | 2838.4 | 5027.2 | 3047 |
| Ant | 2918.25 | 1520 | 4347.5 | **6104.2** | 4415 |

Table 1: Results for best average reward in the first 10M steps of training for our implementations (TRPO (ours) and Trust-PCL) and external implementations. TRPO-GAE are results of Schulman (2017) available on the OpenAI Gym website. TRPO (rllab) and IPG are taken from Gu et al. (2017b). These results are each on different setups with different hyperparameter searches and in some cases different evaluation protocols (*e.g.,* TRPO (rllab) and IPG were run with a simple linear value network instead of the two-hidden layer network we use). Thus, it is not possible to make any definitive claims based on this data. However, we do conclude that our results are overall competitive with state-of-the-art external implementations.

policy) can be competitive in terms of reward while providing a significant improvement in sample efficiency.

One last hyperparameter is $\tau$, determining the degree of exploration. Anecdotally, we found $\tau$ to not be of high importance for the tasks we evaluated. Indeed many of our best results use $\tau = 0$. Including $\tau > 0$ had a marginal effect, at best. The reason for this is likely due to the tasks themselves. Indeed, other works which focus on exploration in continuous control have found the need to propose exploration-advanageous variants of these standard benchmarks (Haarnoja et al., 2017; Houthooft et al., 2016).

## 6 CONCLUSION

We have presented Trust-PCL, an off-policy algorithm employing a relative-entropy penalty to impose a trust region on a maximum reward objective. We found that Trust-PCL can perform well on a set of standard control tasks, improving upon TRPO both in terms of average reward and sample efficiency. Our best results on Trust-PCL are able to maintain the stability and solution quality of TRPO while approaching the sample-efficiency of value-based methods (see *e.g.,* Metz et al. (2017)). This gives hope that the goal of achieving both stability and sample-efficiency without trading-off one for the other is attainable in a single unifying RL algorithm.

## 7 ACKNOWLEDGMENT

We thank Matthew Johnson, Luke Metz, Shane Gu, and the Google Brain team for insightful comments and discussions.

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

## A    IMPLEMENTATION BENEFITS OF TRUST-PCL

We have already highlighted the ability of Trust-PCL to use off-policy data to stably train both a parameterized policy and value estimate, which sets it apart from previous methods. We have also noted the ease with which exploration can be incorporated through the entropy regularizer. We elaborate on several additional benefits of Trust-PCL.

Compared to TRPO, Trust-PCL is much easier to implement. Standard TRPO implementations perform second-order gradient calculations on the KL-divergence to construct a Fisher information matrix (more specifically a vector product with the inverse Fisher information matrix). This yields a vector direction for which a line search is subsequently employed to find the optimal step. Compare this to Trust-PCL which employs simple gradient descent. This makes implementation much more straightforward and easily realizable within standard deep learning frameworks.

Even if one replaces the constraint on the average KL-divergence of TRPO with a simple regularization penalty (as in proximal policy gradient methods (Schulman et al., 2017b; Wang et al., 2016)), optimizing the resulting objective requires computing the gradient of the KL-divergence. In Trust-PCL, there is no such necessity. The per-state KL-divergence need not have an analytically computable gradient. In fact, the KL-divergence need not have a closed form at all. The only requirement of Trust-PCL is that the log-density be analytically computable. This opens up the possible policy parameterizations to a much wider class of functions. While continuous control has traditionally used policies parameterized by unimodal Gaussians, with Trust-PCL the policy can be replaced with something much more expressive—for example, mixtures of Gaussians or autoregressive policies as in Metz et al. (2017).

We have yet to fully explore these additional benefits in this work, but we hope that future investigations can exploit the flexibility and ease of implementation of Trust-PCL to further the progress of RL in continuous control environments.

## B    EXPERIMENTAL SETUP

We describe in detail the experimental setup regarding implementation and hyperparameter search.

### B.1    ENVIRONMENTS

In Acrobot, episodes were cut-off at step $500$. For the remaining environments, episodes were cut-off at step $1,000$.

Acrobot, HalfCheetah, and Swimmer are all non-terminating environments. Thus, for these environments, each episode had equal length and each batch contained the same number of episodes. Hopper, Walker2d, and Ant are environments that can terminate the agent. Thus, for these environments, the batch size throughout training remained constant in terms of steps but not in terms of episodes.

There exists an additional common MuJoCo task called Humanoid. We found that neither our implementation of TRPO nor Trust-PCL could make more than negligible headway on this task, and so omit it from the results. We are aware that TRPO with the addition of GAE and enough fine-tuning can be made to achieve good results on Humanoid (Schulman et al., 2016). We decided to not pursue a GAE implementation to keep a fair comparison between variants. Trust-PCL can also be made to incorporate an analogue to GAE (by maintaining consistencies at varying time scales), but we leave this to future work.

### B.2    IMPLEMENTATION DETAILS

We use fully-connected feed-forward neural networks to represent both policy and value.

The policy $\pi_\theta$ is represented by a neural network with two hidden layers of dimension $64$ with $\tanh$ activations. At time step $t$, the network is given the observation $s_t$. It produces a vector $\mu_t$, which is combined with a learnable (but $t$-agnostic) parameter $\xi$ to parametrize a unimodal Gaussian with mean $\mu_t$ and standard deviation $\exp(\xi)$. The next action $a_t$ is sampled randomly from this Gaussian.

The value network $V_\phi$ is represented by a neural network with two hidden layers of dimension $64$ with $\tanh$ activations. At time step $t$ the network is given the observation $s_t$ and the component-wise squared observation $s_t \odot s_t$. It produces a single scalar value.

### B.2.1 TRPO LEARNING

At each training iteration, both the policy and value parameters are updated. The policy is trained by performing a trust region step according to the procedure described in Schulman et al. (2015).

The value parameters at each step are solved using an LBFGS optimizer. To avoid instability, the value parameters are solved to fit a mixture of the empirical values and the expected values. That is, we determine $\phi$ to minimize $\sum_{s\in\text{batch}}(V_\phi(s) - \kappa V_{\tilde\phi}(s) - (1-\kappa)\hat{V}_{\tilde\phi}(s))^2$, where again $\tilde\phi$ is the previous value parameterization. We use $\kappa = 0.9$. This method for training $\phi$ is according to that used in Schulman (2017).

### B.2.2 TRUST-PCL LEARNING

At each training iteration, both the policy and value parameters are updated. The specific updates are slightly different between Trust-PCL (on-policy) and Trust-PCL (off-policy).

For Trust-PCL (on-policy), the policy is trained by taking a single gradient step using the Adam optimizer (Kingma & Ba, 2015) with learning rate $0.001$. The value network update is inspired by that used in TRPO we perform 5 gradients steps with learning rate $0.001$, calculated with regards to a mix between the empirical values and the expected values according to the previous $\tilde\phi$. We use $\kappa = 0.95$.

For Trust-PCL (off-policy), both the policy and value parameters are updated in a single step using the Adam optimizer with learning rate $0.0001$. For this variant, we also utilize a target value network (lagged at the same rate as the target policy network) to replace the value estimate at the final state for each path. We do not mix between empirical and expected values.

### B.3 HYPERPARAMETER SEARCH

We found the most crucial hyperparameters for effective learning in both TRPO and Trust-PCL to be $\epsilon$ (the constraint defining the size of the trust region) and $d$ (the rollout determining how to evaluate the empirical value of a state). For TRPO we performed a grid search over $\epsilon \in \{0.01, 0.02, 0.05, 0.1\}, d \in \{10, 50\}$. For Trust-PCL we performed a grid search over $\epsilon \in \{0.001, 0.002, 0.005, 0.01\}, d \in \{10, 50\}$. For Trust-PCL we also experimented with the value of $\tau$, either keeping it at a constant $0$ (thus, no exploration) or decaying it from $0.1$ to $0.0$ by a smoothed exponential rate of $0.1$ every $2,500$ training iterations.

We fix the discount to $\gamma = 0.995$ for all environments.

## C PSEUDOCODE

A simplified pseudocode for Trust-PCL is presented in Algorithm 1.

---

**Algorithm 1** Trust-PCL

---

**Input:** Environment $ENV$, trust region constraint $\epsilon$, learning rates $\eta_\pi, \eta_v$, discount factor $\gamma$, rollout $d$, batch size $Q$, collect steps per train step $P$, number of training steps $N$, replay buffer $RB$ with exponential lag $\beta$, lag on prior policy $\alpha$.

**function** Gradients($\{s_{t:t+P}^{(k)}\}_{k=1}^B$)
    *// $C$ is the consistency error defined in Equation 22.*
    Compute $\Delta\theta = \sum_{k=1}^B \sum_{p=0}^{P-1} C(s_{t+p:t+p+d}^{(k)}, \theta, \phi)\nabla_\theta C(s_{t+p:t+p+d}^{(k)}, \theta, \phi)$.
    Compute $\Delta\phi = \sum_{k=1}^B \sum_{p=0}^{P-1} C(s_{t+p:t+p+d}^{(k)}, \theta, \phi)\nabla_\phi C(s_{t+p:t+p+d}^{(k)}, \theta, \phi)$.
    *Return $\Delta\theta, \Delta\phi$*
**end function**

Initialize $\theta, \phi, \lambda$, set $\tilde{\theta} = \theta$.
Initialize empty replay buffer $RB(\beta)$.
**for** $i = 0$ **to** $N - 1$ **do**
    *// Collect*
    Sample $P$ steps $s_{t:t+P} \sim \pi_\theta$ on $ENV$.
    Insert $s_{t:t+P}$ to $RB$.

    *// Train*
    Sample batch $\{s_{t:t+P}^{(k)}\}_{k=1}^B$ from $RB$ to contain a total of $Q$ transitions ($B \approx Q/P$).
    $\Delta\theta, \Delta\phi = $ Gradients($\{s_{t:t+P}^{(k)}\}_{k=1}^B$).
    Update $\theta \leftarrow \theta - \eta_\pi\Delta\theta$.
    Update $\phi \leftarrow \phi - \eta_v\Delta\phi$.

    *// Update auxiliary variables*
    Update $\tilde{\theta} = \alpha\tilde{\theta} + (1 - \alpha)\theta$.
    Update $\lambda$ in terms of $\epsilon$ according to Section 4.3.
**end for**

---

