# OpenReview forum: "Trust-PCL: An Off-Policy Trust Region Method for Continuous Control"
_ICLR.cc/2018/Conference — Accept (Poster)_

### Official Review · AnonReviewer3 · 2017-11-22
**It might be useful but looks like an incremental work. The technical presentation is not quite clear.**

**Rating:** 5
**Confidence:** 1

**Review:**

The paper extends softmax consistency by adding in a relative entropy term to the entropy regularization and applying trust region policy optimization instead of gradient descent.  I am not an expert in this area. It is hard to judge the significance of this extension.

The paper largely follows the work of Nachum et al 2017. The differences (i.e., the claimed novelty) from that work are the relative entropy and trust region method for training. However, the relative entropy term added seems like a marginal modification. Authors claimed that it satisfies the multi-step path consistency but the derivation is missing.

I am a bit confused about the way trust region method is used in the paper. Initially,  problem is written as a constrained optimization problem (12). It is then converted into a penalty form for softmax consistency. Finally, the Lagrange parameter is estimated from the trust region method. In addition, how do you get the Lagrange parameter from epsilon?

The pseudo code of the algorithm is missing. It would be much clearer if a detailed description of the algorithmic procedure is given.

How is the performance of Trust-PCL compared to PCL?

---

> ### Author Response · Authors · 2018-01-05
> **Response**
>
> R3: "The paper largely follows the work of Nachum et al 2017. The differences (i.e., the claimed novelty) from that work are the relative entropy and trust region method for training. However, the relative entropy term added seems like a marginal modification."
>
> The extension of the work of Nachum et al. by including relative entropy is novel and significant because it enables applying softmax consistency to difficult continuous control tasks. Nachum et al (2017) only evaluated PCL on simple discrete control tasks, and without including the additional trust region term, we were not able to obtain promising results. Our results achieve state-of-the-art in continuous control by substantially outperforming TRPO. Other than the introduction of relative entropy as an implicit trust region constraint, the technique described in Section 4.3 is novel and plays a key role in the success of Trust-PCL.
>
> R3: "Authors claimed that it satisfies the multi-step path consistency but the derivation is missing."
>
> We apologize for the lack of clarity. We have updated the paper to expand the derivation of the multi-step consistency over several equations (see Eqs. 16-21).
>
> R3: "I am a bit confused about the way trust region method is used in the paper. Initially,  problem is written as a constrained optimization problem (12). It is then converted into a penalty form for softmax consistency. Finally, the Lagrange parameter is estimated from the trust region method. In addition, how do you get the Lagrange parameter from epsilon?"
>
> Trust-PCL trains towards a trust region objective (Eq. 12 or equivalently Eq. 14) implicity by training a policy and a value function to satisfy a set of path-wise consistencies on off-policy data (Eq. 21). The Lagrange multiplier \lambda is easier to work with to formulate the path-wise consistencies, but \lambda is not constant for a fixed \epsilon, and \epsilon is easier and more intuitive to tune. Hence, we describe a technique in Section 4.3 to adjust \lambda for a given \epsilon, and in the paper we switch between the constraint and Lagrangian form.
>
> R3: "The pseudo code of the algorithm is missing. It would be much clearer if a detailed description of the algorithmic procedure is given."
>
> Good suggestion. We have updated the paper to include a pseudo code of the algorithm in Appendix C. The link to the source code will become available after the blind review as well (footnote 1).
>
> R3: "How is the performance of Trust-PCL compared to PCL?”
>
> PCL is equivalent to Trust-PCL with \epsilon = infinity or \lambda = 0. Section 5.2.1 shows the effect of different values of \epsilon on the results of Trust-PCL. It is clear that as \epsilon increases, the solution quality of Trust-PCL quickly degrades. We found that PCL (corresponding to an even larger \epsilon) is largely ineffective on the difficult continuous control tasks considered in the paper. This shows the significance of the new technique over the original PCL.

---

### Official Review · AnonReviewer1 · 2017-11-24
**Good paper**

**Rating:** 6
**Confidence:** 4

**Review:**

Clarity
The paper is well-written and clear.

Originality
The paper proposes a path consistency learning method with a new combination of entropy regularization and relative entropy. The paper leverages a novel method in determining the coefficient of relative entropy.

Significance
- Trust-PCL achieves overall competitive with state-of-the-art external implementations.
- Trust-PCL (off-policy) significantly outperform TRPO in terms of data efficiency and final performance.
- Even though the paper claims Trust-PCL (on-policy) is close to TRPO, the initial performance of TRPO looks better in HalfCheetah, Hopper, Walker2d and Ant.
- Some ablation studies (e.g., on entropy regularization and relative entropy) and sensitivity analysis on parameters (e.g. \alpha and update frequency on \phi) would be helpful.

Pros:
- The paper is well-written and clear.
- Competitive with state-of-the-art external implementations
- Significant empirical advantage over TRPO.
-  Open source codes.

Cons:
- No ablation studies.

---

> ### Author Response · Authors · 2018-01-05
> **Response**
>
> We thank the reviewer for carefully reading the details of the paper; we greatly appreciate it.
>
> R1: "Even though the paper claims Trust-PCL (on-policy) is close to TRPO, the initial performance of TRPO looks better in HalfCheetah, Hopper, Walker2d and Ant."
>
> Trust-PCL (on-policy) achieves equal or better final reward compared to TRPO, but TRPO has a better initial performance. The results of Trust-PCL (off-policy) are the main point of the paper, showing that we can get both stability and sample-efficiency at the same time in a single algorithm. The presentation of the results for Trust-PCL (on-policy) is to convey the advantage of using off-policy data.
>
> R1: "Some ablation studies (e.g., on entropy regularization and relative entropy) and sensitivity analysis on parameters (e.g. \alpha and update frequency on \phi) would be helpful."
> Section 5.2.1 of the the paper shows the effect of changing \epsilon on the performance. As discussed in Section 4.3, the value of \epsilon directly determines \lambda, the coefficient of relative entropy. The main contribution of the paper is stabilizing off-policy training via a suitable trust region constraint and hence, \epsilon and \lambda are the key hyper-parameters. However, we have expanded Section 5.2.1 to include anecdotal experience regarding the values of \tau and the degree of off/on-policy (determined by \beta, \alpha, P).

---

### Official Review · AnonReviewer2 · 2017-12-04
**Technique is not clear. Contribution is more like incremental. Experiments are insufficient.**

**Rating:** 5
**Confidence:** 4

**Review:**

This paper presents a policy gradient method that employs entropy regularization and entropy constraint at the same time. The entropy regularization on action probability is to encourage the exploration of the policy, while the entropy constraint is to stabilize the gradient.

The major weakness of this paper is the unclear presentation. For example, the algorithm is never fully described, though a handful variants are discussed. How the off-policy version is implemented is missing.

In experiments, why the off-policy version of TRPO is not compared. Comparing the on-policy results, PCL does not show a significant advantage over TRPO. Moreover, the curves of TRPO is so unstable, which is a bit uncommon.

What is the exploration strategy in the experiments? I guess it was softmax probability. However, in many cases, softmax does not perform a good exploration, even if the entropy regularization is added.

Another issue is the discussion of the entropy regularization in the objective function. This regularization, while helping exploration, do changes the original objective. When a policy is required to pass through a very narrow tunnel of states, the regularization that forces a wide action distribution could not have a good performance. Thus it would be more interesting to see experiments on more complex benchmark problems like humanoids.

---

> ### Author Response · Authors · 2018-01-05
> **Response**
>
> R2: "This paper presents a policy gradient method that employs entropy regularization and entropy constraint at the same time… "
>
> Our paper does not present a policy gradient method.  Rather, we show that the optimal policy for an expected reward objective regularized with entropy and relative entropy satisfies a set of path-wise consistencies. Then, we propose an off-policy algorithm to implicitly train towards this objective.
>
> R2: "The major weakness of this paper is the unclear presentation. For example, the algorithm is never fully described, though a handful variants are discussed. How the off-policy version is implemented is missing."
>
> To improve the clarity of the presentation, we have updated the paper and included a pseudo-code in Appendix C. Moreover, we included the implementation details in Appendix B, and we have released an open-source package with all of the variants of the algorithm for completeness (see footnote 1; the link will become available after the blind review).
>
> R2: "In experiments, why the off-policy version of TRPO is not compared."
>
> Unfortunately, TRPO is restricted to the use of on-policy data. This is the major limitation of TRPO. We address this limitation by introducing Trust-PCL, which optimizes a trust region objective using off-policy data. This is the major contribution of the paper
>
> R2: "Comparing the on-policy results, PCL does not show a significant advantage over TRPO."
>
> The results of Trust-PCL (off-policy) are the key takeaway of the paper, showing that we obtain both stability and sample-efficiency in a single algorithm, significantly outperforming TRPO. We present the results of Trust-PCL (on-policy) for completeness, to give a curious reader a sense of the performance loss when only on-policy data is used. We expect practitioners to only use off-policy Trust-PCL.
>
> R2: "the curves of TRPO is so unstable, which is a bit uncommon."
>
> Our TRPO implementation obtains similar performance compared with other implementations by J Schulman and rllab. In Table 1, we compare against a number of externally available implementations.  We also find the stability of our TRPO curves to be qualitatively similar to those appearing externally.
>
> R2: "What is the exploration strategy in the experiments?"
>
> All of the algorithms in the paper have a model of the policy \pi_\theta during training (parameterized as a unimodal Gaussian, as is standard for continuous control).  Accordingly, this policy is used to sample actions. Thus, there is no additional exploration injected. This is standard for continuous control RL algorithms like TRPO.
>
> R2: "I guess it was softmax probability. However, in many cases, softmax does not perform a good exploration, even if the entropy regularization is added."
>
> Please note that the multinomial distribution (so-called softmax probability) is standard in *discrete* control to parametrize the policy, but we are mostly considering continuous control problems in this paper. Our policy is parameterized by a unimodal Gaussian, as is standard in the continuous control benchmarks we evaluate.
>
> R2: "Another issue is the discussion of the entropy regularization in the objective function. This regularization, while helping exploration, do changes the original objective. When a policy is required to pass through a very narrow tunnel of states, the regularization that forces a wide action distribution could not have a good performance."
>
> Augmenting the expected reward objective using entropy regularization is standard in reinforcement learning. Often the multiplier of entropy is annealed to zero by the end of training to enable learning concentrated policies.
>
> R2: "Thus it would be more interesting to see experiments on more complex benchmark problems like humanoids."
>
> We included 6 standard benchmarks in the paper, including: Acrobot, Half Cheetah, Swimmer, Hopper, Walker2D, and Ant. On all of the environments our Trust-PCL (off-policy) algorithm outperforms TRPO in both final reward and sample efficiency. We believe these experiments are enough to demonstrate the promise of the approach.

---

> > ### Comment · AnonReviewer2 · 2018-01-12
> > **Improved. But still not good enough.**
> >
> > The revised paper has made improvements. I thus raise my score a bit. However there are still some issues:
> >
> > "Our paper does not present a policy gradient method" <- This is obviously untrue.
> >
> > "Unfortunately, TRPO is restricted to the use of on-policy data" <- There is no such restrictions. Actually, the proposed Trust-PCL does NOT deal with off-policy data, but only that the authors believe that it can handle off-policy data and thus feed it with such data. This is the same for TRPO. Off-policy data can also be fed to TRPO to see how well TRPO works, which is a must baseline.
> >
> > "We included 6 standard benchmarks in the paper" <- They are the simplest ones.
> >
> > The major concern is the unclear relationship between the methodology of entropy regularization and entropy constraint and the goal of off-policy learning.

---

> > > ### Author Response · Authors · 2018-01-14
> > > **Clarifications**
> > >
> > > These comments continue to reveal some fundamental misunderstandings we should clarify.
> > >
> > > R2: "Our paper does not present a policy gradient method" <- This is obviously untrue.
> > >
> > > - To correct such a misunderstanding, one first needs to realize that policy gradient algorithms update model parameters along the gradient that maximizes expected reward (or possibly a regularized variant thereof).  By contrast, the Trust-PCL updates are not designed to follow any gradient that maximizes (even regularized) expected reward.  Instead, they are designed to minimize a temporal consistency error, much like Q-learning algorithms.
> > >
> > > - If attempting to suggest that any algorithm that updates a parametric policy representation is automatically a “policy gradient” method, that would not be consistent with the standard terminology used in RL.
> > >
> > >
> > > R2: "Unfortunately, TRPO is restricted to the use of on-policy data" <- There is no such restrictions. Actually, the proposed Trust-PCL does NOT deal with off-policy data, but only that the authors believe that it can handle off-policy data and thus feed it with such data. This is the same for TRPO. Off-policy data can also be fed to TRPO to see how well TRPO works, which is a must baseline.
> > >
> > > - It is again helpful to understand that Trust-PCL is off-policy in the same sense that Q-learning is off-policy: both use off-policy data to train models to satisfy temporal consistencies.  Models learned in this way then induce a policy that is optimal iff all the temporal consistencies are satisfied.
> > >
> > > - The consistencies expressed in this paper (Eq. 21) are off-policy consistencies, in the sense that they do not rely on a specific sampling distribution of actions.  Note that the expectations are only with respect to environment stochasticity (which is covered by a replay buffer).  If the environment were deterministic, the consistency equations would still be well posed and contain no expectations.
> > >
> > > - Other work has also used similar ideas to derive off-policy algorithms:
> > > - -- Nachum, et al. 2017: “Bridging the Gap Between Value and Policy Based Reinforcement Learning”
> > > - -- Peters, et al. 2010: “Relative Entropy Policy Search”
> > >
> > > - TRPO updates involve importance weights of the form [current policy action probability] / [behavior policy action probability].  For this reason, off-policy training becomes highly unstable if the current policy deviates too far from the behavior policy.  We are not aware of any work that trains TRPO in an off-policy manner using a replay buffer.  The naive suggestion given above obviously leads to wild instability.  If the reviewer is aware of any successful such attempts, a reference would certainly be appreciated, if any exists.
> > >
> > > - One could perhaps attempt to interpret the learning rate of TRPO (\epsilon, the max divergence) as a way to tune the degree of “on-policy” behavior.  Under such an interpretation one could then attempt to choose \epsilon to be as large as possible without incurring instability.  However, our experiments have already demonstrated the results of using the best \epsilon after exhaustive tuning, so there would not be any additional information to be gained through such an interpretation.
> > >
> > >
> > > R2: "We included 6 standard benchmarks in the paper" <- They are the simplest ones.
> > >
> > > - The tasks we included cover a wide range of difficulties.  The results show significant advantages on harder tasks such as Hopper, Walker2d, and Ant.  These tasks are by no means “simple” as can be deduced by comparing our results to those in other papers, including many submitted to this year’s ICLR:
> > > - -- https://openreview.net/forum?id=H1tSsb-AW
> > > - -- https://openreview.net/forum?id=BkUp6GZRW
> > > - -- https://openreview.net/forum?id=HJjvxl-Cb
> > > - -- https://openreview.net/forum?id=B1nLkl-0Z

---

### Decision · Program_Chairs · 2018-01-29
**ICLR 2018 Conference Acceptance Decision**

**Decision:**

Accept (Poster)

**Comment:**

This paper adapts (Nachum et al 2017) to continuous control via TRPO.   The work is incremental (not in the dirty sense of the word popular amongst researchers, but rather in the sense of "building atop a closely related work"), nontrivial,  and shows empirical promise.    The reviewers would like more exploration of the sensitivity of the hyper-parameters.